# Rapidly receding Arctic Canada glaciers revealing landscapes continuously ice-covered for more than 40,000 years

Simon L. Pendleton [1], Gifford H. Miller[1], Nathaniel Lifton [2,3], Scott J. Lehman[1], John Southon[4], Sarah E. Crump [1] & Robert S. Anderson[1]

Arctic temperatures are increasing faster than the Northern Hemisphere average due to strong positive feedbacks unique to polar regions. However, the degree to which recent Arctic warming is unprecedented remains debated. Ages of entombed plants in growth position preserved by now receding ice caps in Arctic Canada help to address this issue by placing recent conditions in a multi-millennial context. Here we show that pre-Holocene radiocarbon dates on plants collected at the margins of 30 ice caps in Arctic Canada suggest those locations were continuously ice covered for > 40 kyr, but are now ice-free. We use in situ [14]C inventories in rocks from nine locations to explore the possibility of brief exposure during the warm early Holocene. Modeling the evolution of in situ [14]C confirms that Holocene exposure is unlikely at all but one of the sites. Viewed in the context of temperature records from Greenland ice cores, our results suggest that summer warmth of the past century exceeds now any century in ~115,000 years.

[1] INSTAAR and Department of Geological Sciences, University of Colorado, Boulder, CO 80309-0450, USA. [2] Department of Earth, Atmospheric, and Planetary Sciences, Purdue University, West Lafayette, IN 47907, USA. [3] Department of Physics and Astronomy, PRIME Lab, Purdue University, West Lafayette, IN 47907, USA. [4] Department of Earth System Science, University of California, Irvine, Croul Hall, Irvine, CA 92697-3100, USA. Correspondence and requests for materials should be addressed to S.L.P. (email: simon.pendleton@colorado.edu)

Summer warming in the Arctic has significantly outpaced global average warming over the past several decades[1,2] and Arctic glaciers are consequently retreating rapidly, especially in the Eastern Canadian Arctic[3], where summer air temperature explains >90% of the year-to-year variability in glacier surface mass balance[4]. The large influence of summer temperature on glacier mass balance also suggests that past changes in glacier dimensions can be used to place contemporary warming in a multi-millennial perspective[5,6]. Here we exploit the high potential for thin, cold-based, non-erosive ice caps to preserve the landscapes over which they advance, including delicate tundra plants and un-eroded rocks. Under these conditions, radiocarbon dates of plants newly exposed by recent melting (hereafter plant [14]C) constrain the time of plant death due to snowline lowering and glacier expansion[7–9]. Recently reported plant [14]C dates > 40 ka from four ice caps on Baffin Island, Arctic Canada, further suggest that ice cover at those locations may have persisted through the warmest part of the Holocene[7]. However, given the small number of sites, it is possible that local conditions may have favored anomalous preservation of plants following melt out and exposure, allowing for subsequent reburial. Here we expand the number of ice caps with plant [14]C ages > 40 ka from 4 to 30, greatly reducing the likelihood that the earlier dates reflected anomalous conditions. We also provide in situ cosmogenic [14]C inventories from rock surfaces ([14]C produced within minerals via cosmic ray bombardment; hereafter in situ [14]C) adjacent to plant collections with radiocarbon ages > 40 ka to test the assertion that such sites were continuously covered by ice throughout the Holocene. The combination of 30 retreating ice caps revealing plants with pre-Holocene radiocarbon ages and measured in situ [14]C inventories provides strong evidence for continuous ice coverage of these locations for >40 ka. When viewed in context of temperature records from Greenland ice cores, these results suggest that the past century of warming is likely greater than any preceding century in the past ~115,000 years.

## Results

**Newly exposed plants killed by snowline lowering >40 ka.** Forty-eight in situ tundra plants collected within 1 m of the ice margin at the time of collection from 30 different ice caps on eastern Baffin Island (Fig. 1) were dated through accelerator mass spectrometry (AMS) [14]C analysis; with radiocarbon ages of 40 to >50 ka, close to or beyond the range of [14]C dating (Table 1). Most of the sampled ice caps are small (1–2 km²), and all lie within a region of 170 × 70 km (Fig. 1). Replicate plant collections were [14]C dated at nine ice caps, including separate strands from the same plant and different plants collected within 100 m along the same ice margin. Some early collections that did not undergo rigorous pretreatment because they were expected to be < 10 ka returned apparent finite ages < 40 ka ([14]C age), but subsequent analyses of the same samples following rigorous pretreatment yielded [14]C ages > 40 ka (sites 11 and 12; Table 1).

Plant [14]C ages define the time when summer temperature decline resulted in snowline lowering, leading directly to permanent plant burial by snow or by ice margin expansion across the site shortly thereafter. Prior field observations suggest that once plants are exposed by ice recession, they are efficiently removed from the landscape by meltwater in summer and wind-blown snow in winter[7]. In addition, colonization by new plants, and in some instances regrowth of recently exposed dead moss, begins within 1–3 years[10]. The combination of rapid removal and high rate of contemporary ice retreat (0.5–1.0 m yr⁻¹ vertical lowering of glacier surfaces[3], corresponding to ~10 m yr⁻¹ rate of horizontal retreat in most settings), suggests that plants collected

within 1 m of the ice margin were likely first exposed the year they were collected. In rare cases, preservation of dead plants exposed by ice retreat has been noted up to 200 m beyond current ice margins, indicating that they have survived on the landscape for several decades following exposure[11,12]. Given the possibility that some plants may have been exposed and then re-entombed, we independently evaluate the exposure history of the sites using in situ [14]C in associated rock samples.

**In situ [14]C inventories.** The in situ [14]C inventory in the surface of rocks now emerging from beneath receding cold-based ice reflects the cumulative [14]C production during exposure when the rocks were ice-free, and attenuated production when rocks were covered by thin ice, as well as losses due to the continuous decay of [14]C (half-life 5700 ± 30 years[13]) and erosion of the rock surface. Unlike plant [14]C, which records the timing of plant death by snow and/or ice cover, in situ [14]C records the cumulative exposure and burial history of a rock surface over the lifespan of the isotope (i.e., several half-lives). Two mechanisms dominate in situ [14]C production: (1) energetic spallation reactions between a high-energy nucleon and a target atom, and (2) low-energy (slow) muogenic reactions that occur when a negatively charged muon falls into the electron shell of an atom and is captured by the nucleus[14]. Various higher-energy (fast) muogenic reactions are less important for in situ [14]C production[15–17] but cannot be neglected. Although in situ [14]C production is dominated by spallation in unshielded rock surfaces[15–17], these reactions are negligible beneath ~10 m of ice[18]. Muogenic production in unshielded rock surfaces is much lower than production by spallation, but decreases more slowly with ice thickness, not reaching negligible amounts until ice thickness exceeds 40 m. Since any in situ [14]C produced prior to ~40 ka will have decayed below detection, and all sites are expected to have been covered by at least 40 m of ice during the last glacial maximum (LGM), in situ [14]C inventories[19–21] at all our sites must have been negligible by the onset of deglaciation. Thus, measured in situ [14]C

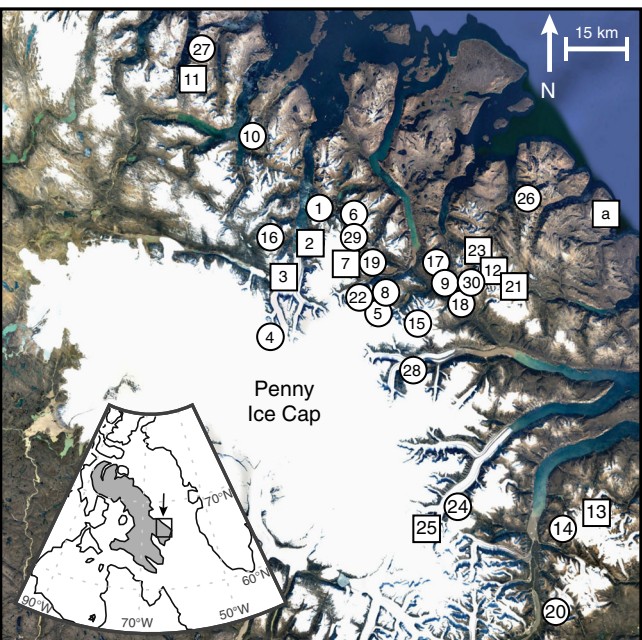

**Fig. 1** Map showing sample localities on eastern Baffin Island. White circles indicate locations of plant samples, squares indicate locations with both plant and rock (in situ cosmogenic [14]C) samples. Site a is an unglaciated steep-sided summit where only rock was sampled (imagery: Google Earth: Image IBCAO, Landsat/Copernicus)

### Table 1 Sample locations and $^{14}$C ages

| Site # | Sample ID | $^{14}$C age (yr) | $^{14}$C ± 1$\sigma$ (yr) | Cal age (yr) | ±1$\sigma$ (yr) |
|---|---|---|---|---|---|
| 1 | M13-B002v | >43,300 | – | – | – |
| 2 | M13-B005v | >48,370 | – | – | – |
| 2 | M13-B007v | >45,277 | – | – | – |
| 3 | M13-B011v | 43,770 | 4670 | 45,443 | +4557/−1177 |
| 3 | M14-B101v | >46,320 | – | – | – |
| 4 | M13-B018v | >45,277 | – | – | – |
| 5 | M13-B028v | >45,277 | – | – | – |
| 6 | M13-B045v | >49,990 | – | – | – |
| 7 | M13-B051v | >45,277 | – | – | – |
| 7 | M13-B052v | >47,000 | – | – | – |
| 7 | M14-B139v | >44,940 | – | – | – |
| 8 | M13-B055v | >45,277 | – | – | – |
| 9 | M13-B064v | 41,800 | 3250 | 45,171 | +2893/−2420 |
| 10 | M13-B066v | 45,830 | 1770 | 48,199 | +1801/-520 |
| 10 | M13-B069v | >47,800 | – | – | – |
| 11 | M13-B094v | 48,850 | 2570 | 48,491 | +1509/−390 |
| 11 | M13-B091v | 45,240 | 2570 | 47,449 | +2551/−730 |
| 11[ab] | M10-B258v | 34,300 | 3600 | 38,214 | +3662/−3200 |
| 11[ab] | M10-B258v | 39,740 | 950 | 43,550 | +689/−810 |
| 11[ab] | M10-B258v | 37,510 | 490 | 41,880 | +380/−320 |
| 12 | M13-B104v | >45,277 | – | – | – |
| 12[ab] | M10-B231v | 29,100 | 1500 | 33,094 | +1265/−1600 |
| 12[ab] | M10-B231v | 44,300 | 1300 | 47,570 | +1306/−1280 |
| 12[ab] | M10-B231v | 23,920 | 100 | 27,959 | +97/−150 |
| 12[ab] | M10-B232v | 37,500 | 3600 | 41,194 | +3738/−3110 |
| 13 | M13-B195v | 52,120 | 3860 | 48,226 | +1774/−460 |
| 13 | M13-B196v | 42,100 | 1270 | 45,545 | +1080/-1280 |
| 14 | M13-B201v | 50,300 | 3080 | 48,419 | +1581/−410 |
| 15 | M14-B020v | >45,650 | – | – | – |
| 16 | M14-B085v | 39,280 | 1230 | 43,228 | +910/-980 |
| 17 | M14-B107v | >46,320 | – | – | – |
| 18 | M14-B113v | >46,320 | – | – | – |
| 19 | M14-B143v | >46,320 | – | – | – |
| 20 | M14-B154v | >45,980 | – | – | – |
| 21 | M14-B158v | >46,320 | – | – | – |
| 22 | M14-B163v | >46,380 | – | – | – |
| 23 | M14-B164v | >45,220 | – | – | – |
| 23 | M14-B165V | 46,120 | 2870 | 47,592 | +2408/-690 |
| 24 | M14-B183v | 45,780 | 2750 | 47,549 | +2451/-700 |
| 24 | M14-B184v | >46,320 | – | – | – |
| 25 | M15-B047v | >47,000 | – | – | – |
| 25 | M15-B048v | >44,400 | – | – | – |
| 26[ab] | M10-B247v | 45,600 | 2500 | 47,636 | +2364/−680 |
| 27[ab] | M10-B255v | 43,200 | 2700 | 46,338 | +2541/−1830 |
| 27[ab] | M10-B256v | 50,700 | 3100 | 48,468 | +1532/−390 |
| 28 | M14-B009v | 44,200 | 1850 | 47,303 | +1842/−1370 |
| 29 | M13-B046v | >50,143 | – | – | — |
| 30 | M14-B161v | >50,768 | – | – | – |

All plant samples were collected between 2010 and 2015 (year of collection denoted by sample ID prefix, M10-, M13-, etc.). Samples with > are minimum limiting ages and indistinguishable from the organic measurement blank. All other samples are also reported in calibrated years BP using IntCal 2013 and OxCal 4.2.4[50,51]. For sample metadata see Supplementary Table 1
[a]From Miller et al.[7]
[b]Received only deionized water pretreatment

inventories reflect primarily the local variation of ice thickness history during and after regional deglaciation.

Measured in situ $^{14}$C concentrations in rock surfaces adjacent to nine of the study sites with plant $^{14}$C ages > 40 ka (Fig. 1) are all above established detection limits (Supplementary Table 2)[19] and range between 7000 and 133,400 at g$^{-1}$, with samples from all but two sites below 73,000 at g$^{-1}$ (Table 2, Fig. 2, Supplementary Fig. 1). In contrast, a nearby unglaciated, steep-sided coastal summit (site a, Fig. 1) that was likely never glaciated, has an in situ $^{14}$C inventory of ~368,800 at g$^{-1}$ (Table 2), consistent with continuous subaerial exposure for more than 20 ka. Farther north on Baffin Island, an in situ $^{14}$C concentration of ~249,000 at g$^{-1}$ was reported in rock at 939 m above sea level (asl) from a location that has been ice-free since local deglaciation ~13 ka[22]. Several rocks on the central Baffin Island plateau repeatedly exposed and buried by local ice caps during the Holocene have in situ $^{14}$C concentrations of ~85,000 at g$^{-1}$ [23]. The new inventories reported here are generally significantly less than reported elsewhere on Baffin Island, despite site-specific production rates at the new, higher elevation sites that are on average ~60% higher than at these other locations on Baffin Island (Supplementary Table 3). These comparisons indicate that most of the newly sampled sites likely experienced significantly more burial than the other Baffin Island sites.

**Ice cover simulations**. We use a numerical simulation to estimate in situ $^{14}$C inventories for a range of plausible post-LGM ice

**Table 2 In situ $^{14}$C concentrations and ice cover simulations**

| Site # | Elevation (m) | $^{14}$C conc. (at g$^{-1}$) | ± (at g$^{-1}$) | Max. exposure prior to 4.5 ka (kyr) | Continuous Holocene coverage | |
|---|---|---|---|---|---|---|
| | | | | | Holocene ice thickness (m) | Peak LIA ice thickness (m) |
| 2 | 1478 | 7000 | 1500 | 0 | 26 (18–36) | 50 (29–70) |
| 3 | 1478 | 105,800 | 4000 | 0 | 1 (1–2) | 9 (3–11) |
| 7 | 1522 | 65,200 | 3500 | 0 | 2 (2–3) | 18 (5–24) |
| 11 | 1255 | 49,300 | 3500 | 0 | 2 (2–3) | 33 (6–49) |
| 12 | 1389 | 45,700 | 3600 | 0 | 3 (2–4) | 17 (5–70) |
| 13 | 1588 | 20,900 | 3400 | 0 | 6 (4–13) | 29 (10–70) |
| 21 | 1390 | 73,000 | 3700 | 0 | 1 (1–2) | 67 (6–70) |
| 23 | 1436 | 133,400 | 6400 | 5 | 1 (1–1) | 4 (4–4) |
| 25 | 1526 | 18,500 | 3300 | 0 | 7 (4–15) | 29 (11–70) |
| a[a] | 1010 | 368,800 | 7100 | N/A | N/A | N/A |

In situ $^{14}$C concentrations at 9 of the 30 sites with >40 ka plant ages and the results of the Holocene ice cover simulations. Ice thicknesses are reported as the median and range of values consistent with measured in situ $^{14}$C and its analytical error. Sample a is a steep-sided summit that has likely never been glaciated, and so is not included in the ice cover simulations
*LIA* Little Ice Age
[a]Coastal summit bedrock under constant exposure (Holocene ice cover simulations inappropriate), reported for comparison.

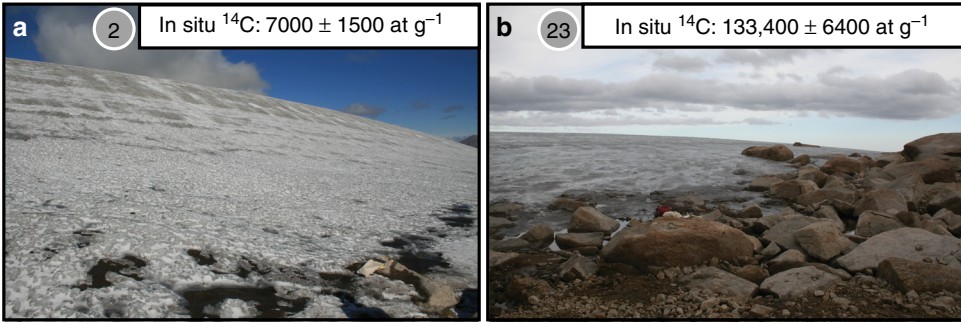

**Fig. 2** In situ $^{14}$C sample sites. Photographs of the **a** lowest (site #2) and **b** highest (site #23) inventories of in situ $^{14}$C sample locations and measured in situ $^{14}$C concentrations. High levels of in situ $^{14}$C production through the thin ice at site #23 relative to thick ice at site #2 likely explains the high in situ $^{14}$C inventories at site #23

thickness histories at each site, and compare these with measured inventories. The simulations make use of updated estimates of the dependence of $^{14}$C production on geomagnetic latitude[24,25] and the attenuation lengths for in situ $^{14}$C production from fast and slow muons[17]. Rock erosion is assumed to be negligible given the field evidence that delicate tundra plants are preserved, and the expectation that the ice is well below the pressure melting point and therefore unable to slide. Prior to deglaciation we assume that ice thickness history is a simple linear function of the North Greenland Ice Core Project δ$^{18}$O record[26] of paleotemperature, scaling from zero during the Last Interglacial (LIG) to 70 m during the LGM (a realistic value based on ice rheology and ice cap dimensions[7] as well as field observations). This leads to inventories that are near zero by the onset of deglaciation, a conclusion that is largely insensitive to the functional form of the assumed preceding thickness history (Supplemental Fig. 2). After 12 ka, we assume a local ice thickness history based on the observed and modeled melt and thinning history for the Agassiz Ice Cap, Ellesmere Island[27]; i.e., we impose relatively rapid thinning 12–8 ka from an assumed thickness maximum of 70 m, followed by slower thinning until 4.5 ka, which is a consensus initiation time for Holocene ice regrowth on Baffin Island[7,28]. At 4.5 ka ice thicknesses are specified to increase linearly until the peak of the Little Ice Age (LIA), 1900 CE, before thinning to zero at present. We test for possible ice-free intervals by allowing for exposure in increments of 10 years prior to 4.5 ka for all sample locations. All solutions that are consistent with measured in situ $^{14}$C inventories and their measurement uncertainties are

considered for each site, and expressed in Table 2 as the median and range of possible Holocene and LIA ice thickness.

Under these conditions, only the highest inventory, measured at site 23, permits exposure during the Holocene (Table 2, Supplementary Fig. 3), as its inventory lies above an intrinsic threshold determined by the relationships between exposure, production, and decay. The same constraint prohibits prolonged exposure at the remaining sites. Two other sites with higher inventories (3, 21) have modeled thickness histories that reach a minimum of ~1 m of ice cover during the middle Holocene (Table 2, Fig. 3, Supplementary Fig. 4). While consistent with isotopic constraints, persistence of ice caps of this thickness for long periods is unlikely given natural glacier fluctuation. We note, however, that these are almost certainly minimum thickness estimates that result from our assumption of linear ice growth after 4.5 ka. Any thinning and increased production during that interval would have to be compensated by increased shielding and ice thickness earlier, in order to match measured inventories. Implied minimum ice thicknesses at the remaining sites are all glaciologically plausible. Although our in situ $^{14}$C simulations do not encompass all possible scenarios, they demonstrate that significant Holocene exposure at all but one of the sample sites is unlikely, and is not possible under any realistic scenario for most sample study locations.

**Variable plant $^{14}$C ages revealed by receding ice.** Our results indicating continuous ice cover for at least the past ~40 ka at a large number of sites are not inconsistent with observations from other retreating ice margins in the same region that have revealed

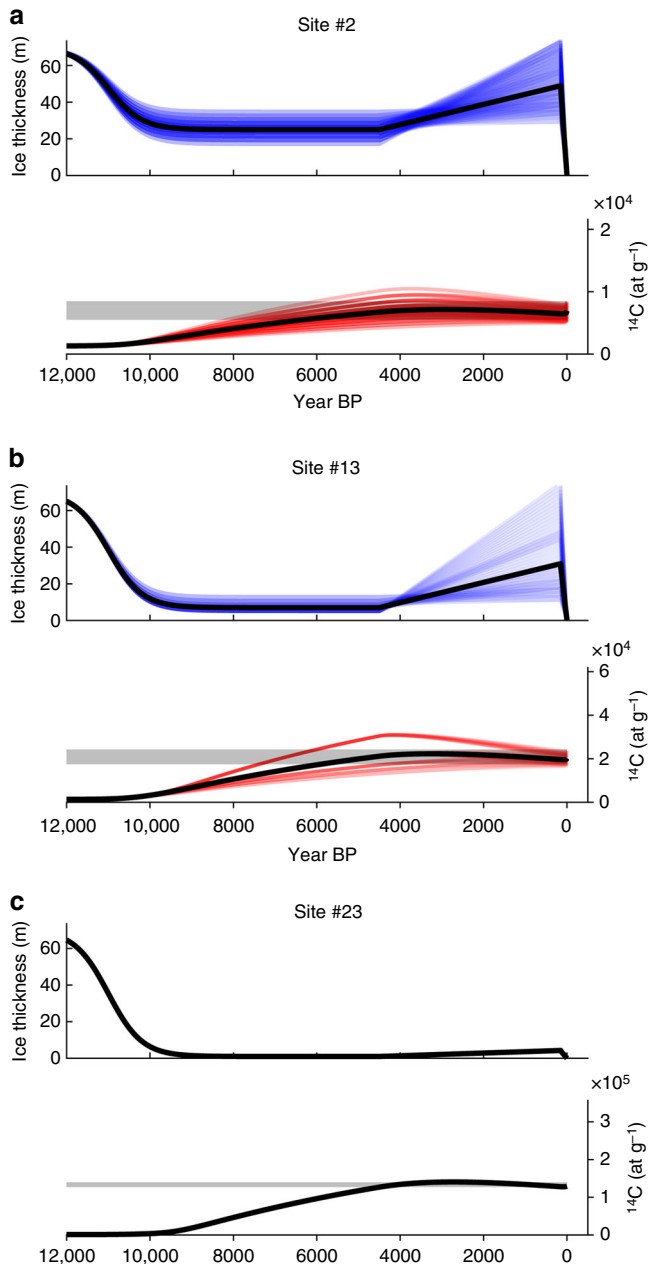

**Fig. 3** Example of ice cover and in situ $^{14}C$ simulations. Model output for the lowest (**a**), intermediate (**b**), and highest (**c**) in situ $^{14}C$ inventory locations showing the range of ice cover histories (blue lines) for the past 12 ka that yield in situ $^{14}C$ concentrations (red lines) within uncertainty of the measured $^{14}C$ (gray bar). Individual simulations are shown as colored lines, median shown as solid black line. See Supplementary Figs. 3 and 4 and Supplementary Table 2 for all simulation results

plants of Holocene age[7,28]. We propose two possible explanations for the range of implied ice cover histories. First, due to differences in the patterns of accumulation and ablation, ice caps rarely expand and recede symmetrically, so that upon retreat, landscapes are not necessarily exposed in uniform reverse chronological order. Second, some ice caps behave as threshold systems[29]. All but 7 of the 30 sites with plant $^{14}C$ > 40 ka are from high elevations, 1380–1600 m asl, and most of these are from the margins of pedestal ice caps (Fig. 4). These ice caps evolve on small (<2 km²), flat-topped summits, surrounded by steep slopes

where maximum ice thicknesses are limited by ice rheology to <70 m[7,30]. Once the equilibrium line altitude (ELA) drops below the pedestal surface, ice will grow quickly to a physically determined thickness, but no thicker, even for continued ELA descent. In contrast, lower elevation ice caps can form and expand on less constrained terrain. Consequently, such ice caps became thicker and covered larger altitudinal ranges than pedestal ice caps that formed much earlier, but with dimensions constrained by topography. The result is that the thickness of pedestal ice caps is insensitive to ELA changes below the pedestal base, but is highly sensitive to ELA change as it rises above that level (Fig. 4).

Warming over the past century has led to a rapid rise of the ELA across our field area to at least 500 m above its LIA minimum[7], resulting in an ELA that is now above the highest ice caps. Consequently, ice caps are melting at all elevations in summer[3]. Melt at higher elevations may have been aggravated by an increase in downward longwave radiation due to increases in water vapor content of the normally dry atmosphere at these high-latitude, high-elevation locations[30–32]. The magnitude of the modern melt rate anomaly is illustrated in the simulations here that show average melt rates over the past 150 years have been five times higher than the average melt rate during the interval of peak summer insolation 11–9 ka. Under these conditions thicker, lower elevation ice caps that formed <5 ka are currently revealing plants with Holocene $^{14}C$ ages, whereas thinner pedestal ice caps are revealing much older landscapes (Fig. 4). Furthermore, some intermediate elevation ice caps that shrank but did not disappear during the Holocene thermal maximum (HTM) are revealing both >40 ka and Holocene landscapes along different margins of the same ice cap (e.g., site #12). Hence the dramatic ELA rise in the Eastern Canadian Arctic over the past century[33] is resulting in the simultaneous exposure of a range of late Holocene (<5 ka[7,28]) and much older (>40 ka) landscapes (Fig. 4).

Enhanced ice retreat due to warming summers has exposed entombed plants, still in growth position, that yield >40 ka plant $^{14}C$ ages from the margins of 30 unique ice caps. These ages constrain the time when ice caps advanced across the sites, implying that they have remained continuously ice-covered until present. We tested this interpretation by measuring in situ cosmogenic $^{14}C$ inventories in adjacent rock surfaces at nine of the sites. Numerical simulations of in situ $^{14}C$ production and removal for a range of realistic ice thickness histories indicate that all but one of the sites must have been ice covered throughout the Holocene. Taken together, the two lines of evidence suggest that many Baffin Island landscapes that have become exposed by ice retreat during recent warming have been continuously ice-covered since at least ~40 ka, including the HTM, when local summer insolation was up to 9% higher than present[34]. Paleotemperature reconstructions from Greenland and Baffin Island show that the most recent time prior to the Holocene with temperatures similar to present was during the Last Interglaciation, suggesting that that these landscapes are now ice-free for the first time in ~115 ka and that modern temperatures represent the warmest century in 115 ka, despite relatively low local summer insolation. These trends are likely to continue and remove all ice from Baffin Island within the next few centuries, even in the absence of additional summer warming[4,35].

## Methods
**Plant $^{14}C$ collection and preparation.** Plant samples were collected late in August, to capitalize on the accessibility of plants exposed by ice margin retreat during the summer melt season. Plants (mostly bryophytes), still in growth position, were collected within 1 m of the modern ice margin at most sites[7]. For this study, one to six strands from each sampled plant (and in rare cases, a single lichen) were sonicated in deionized water and treated with an acid-base-acid[36] wash before being converted to graphite by the Laboratory for AMS Radiocarbon Preparation and Research (NSRL, Univ. of Colorado). Graphite targets were measured at the W.M. Keck

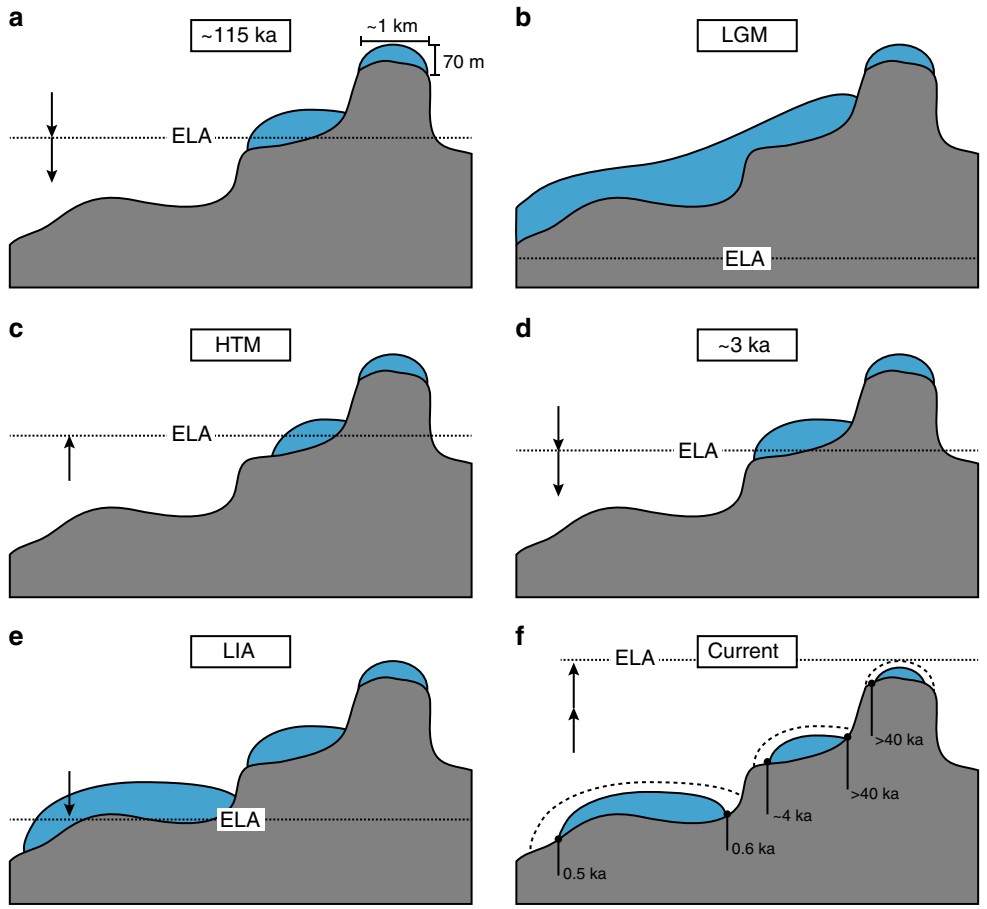

**Fig. 4** Ice cap response to rapid equilibrium line altitude (ELA) rise exposes landscapes of varying ages. Conceptual time series of ELA change illustrating the threshold behavior of pedestal ice caps that reach and maintain their maximum dimensions even when the ELA is relatively high (**a**). During the LGM, much of the landscape is covered with ice but additional growth of pedestal ice caps is limited by topography (**b**). As ELA rises during the deglaciation and early Holocene, many low elevation ice caps disappear, intermediate elevation ice caps recede, whereas pedestal ice caps maintain their size (**c**). Renewed descent of ELA during the Holocene permits lower elevation ice caps to reform (**d**, **e**). The anomalous rise of the ELA over the past century has been so rapid that ice caps are no longer in equilibrium with climate; consequently, the thickest ice caps, often at lower elevation than pedestal ice caps, take longest to disappear, revealing a range of Holocene plant [14]C ages (**f**)

Carbon Cycle Accelerator Mass Spectrometry Laboratory at the University of California Irvine.

**In situ [14]C collection and analysis.** [14]C is produced in situ in quartz from rock surfaces by spallation reactions between incoming secondary nucleons (neutrons + protons) and Si and O. More than ~6 m of ice cover eliminates most in situ [14]C production by spallation. However, muons are only weakly attenuated with depth, allowing low rates of in situ [14]C production via muogenic reactions beneath >6 m of overlying ice. Rocks continuously beneath >6 m of ice since the LIG will have low concentrations of in situ [14]C that reflect muogenic production and decay. Any exposure during the Holocene will greatly increase [14]C inventories by spallation reactions.

Stable granitic rock surfaces located near nine of the plant [14]C collection sites were sampled for in situ [14]C to a maximum depth of 4 c using a hammer and chisel. Quartz was isolated and purified from whole rock at the University of Colorado Boulder Cosmogenic Isotope Laboratory using standard techniques[37]. Samples for in situ [14]C concentration were prepared at the Purdue Rare Isotope Measurement Laboratory (PRIME Lab) using current methods[21]. The automated extraction process involves combusting ~5–10 g of quartz at 600 °C for 1 h to remove atmospheric contaminants, followed by dissolution in a previously degassed LiBO$_2$ flux at 1100 °C for 3 h. Both steps take place in an atmosphere of ca. 50 torr of Research Purity O$_2$. Any carbon species released during the high-temperature step are oxidized to CO$_2$, which is then purified, measured quantitatively, diluted with [14]C-free CO$_2$, and converted to graphite for [14]C measurement by AMS at PRIME Lab[21]. In situ [14]C concentrations were calculated from measured AMS isotope ratios[38], less representative sample-specific procedural blanks.

**In situ [14]C concentrations.** Laboratory procedural blank values reflect an approximately linear decrease between May 5, 2016 and July 29, 2016 from a stable mean value of 401,000 ± 30,000 [14]C atoms to 154,200 ± 30,900 [14]C atoms[21]

(Supplementary Table 2). A linear least squares fit to the upper, lower, and two intermediate values was used to estimate the blanks during the time-dependent period. In all, 30,900 [14]C atoms is a conservative error estimate for all process blanks. M14-B090R was analyzed on a separate extraction system with a corresponding process blank of 160,700 ± 12,100 [14]C atoms. The detection limit is considered to be the uncertainty associated with the process blank measurement[19]. Accounting for the 10 g of quartz dissolved for each sample, all reported concentrations are above established detection limits[21].

**In situ [14]C production simulation.** For each realization of ice thickness history for each sample, we determine the production rate at the rock surface ($P_o$; at g$^{-1}$ yr$^{-1}$) from site elevation, latitude (LSD$n$ scaling framework)[25,39–41], while accounting for attenuation of spallogenic[14,42] and muogenic[15–17,43] production due to ice thickness at each timestep. The in situ [14]C concentrations are iteratively calculated at annual time steps using the following equation[44–49]:

$$N = \left[ N_o * e^{-\lambda dt} \right] + \left[ \frac{P_o}{\lambda} + \left( 1 - e^{-\lambda dt} \right) \right] \qquad (1)$$

where $N_o$ is the existing in situ [14]C concentration (at g$^{-1}$) at the beginning of each timestep, $\lambda$ is the decay constant of [14]C (yr$^{-1}$), and d$t$ is the annual timestep. Differences in production arising from changes in atmospheric density at each site due to sea level rise are negligible at these elevations[44]. Rock erosion is assumed to be negligible given the field evidence that ice cover was cold-based.

## Data availability
The data that support the findings of this study, including MATLAB code used in numerical simulations, are available from the corresponding author upon reasonable request.

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

## Acknowledgements

The authors thank Polar Continental Shelf Project and Universal Helicopters for logistical support, the Inuit of Nunavut for permission to conduct research on their lands, and the Inuit of Qikiqtarjuaq for their hospitality and logistical assistance. This project was supported by NSF awards ARC-1204096 and PLR-1418040. N.L. also acknowledges support from NSF award EAR-1560658. K. Refsnider and C. Florian collected the plants from four of the sites we report. M. Kennedy, D. Sperduto, C. La Farge, N. Young, and P. T. Davis assisted with field collections. The manuscript benefited from insights offered by R.S. Bradley. Publication of this article was funded in part by the University of Colorado Boulder Libraries Open Access Fund.

## Author contributions

S.L.P. and G.H.M. collected the samples. S.L.P. contributed text, laboratory prep, numerical simulations, and figures; G.H.M. and S.J.L. contributed text and expertise to discussion and model setup; N.L. processed in situ [14]C and aided with numerical simulation code; S.J.L. provided plant [14]C ages and contributed insights into radiocarbon age interpretation; J.S. contributed additional plant [14]C ages; S.E.C. and R.S.A. provided text and contributed to interpretations and modeling.

## Additional information

**Competing interests:** The authors declare no competing interests.

