## [Peer Review File · Nature Communications]

Reviewers' comments:

Reviewer #1 (Remarks to the Author):

Review of Pendleton et al., 'Rapidly receding Arctic Canada glaciers revealing landscapes ice-covered for 115,000 years'

The paper describes a glacial-chronologic approach to assessing modern deglaciation in Arctic Canada in the context of past glacial change, and builds on similar work by several of the authors (e.g., Miller et al., 2013). In so doing, the authors present a large radiocarbon dataset and a smaller in situ ^{14}C record, ultimately arriving at the conclusion that ice caps in this part of the Arctic have not been as small as present since the last interglacial, circa 115 ka. Independent of that principal finding, this research represents a large body of work and will make a valuable contribution to the palaeoclimate community's understanding of natural boundary conditions for climate: How fast can our climate system fluctuate between states, and how has temperature evolved over the preceding cold and warm periods? At face value, this paper provides a truly tantalising glimpse of the impacts of abrupt climate shifts.

Methodologically, the plant ^{14}C chronology is a tried and tested (if not particularly novel) approach and the results, showing modern ice extent is unprecedented since at least 40 ka (or the limit of the radiocarbon method), appears to be robust. The subsequent measurement of in situ cosmogenic ^{14}C in bedrock represents a logical progression from the plant data and is indeed a novel element of the paper (more on that below). Moreover, the potential mechanisms by which retreat of these pedestal ice caps might be revealing landscapes today that were buried during the Holocene thermal maximum is well-illustrated in Figure 4. Beyond the plant-based data set, however, I feel the arguments for continuous ice cover since the last interglacial are overstepping the limits of the multi-proxy record, for reasons I hope to outline below.

Firstly, the gap between ~ 40 ka (beyond which the ^{14}C method is unreliable) and the last interglacial constitutes a period of approximately 70,000 years during which climate varied considerably. While the authors might indeed be correct in their inference that ice cover was continuous since 115 ka, there is no way to establish this possibility with a minimum-limiting data set such as this. All they can say with certainty is that ice-free conditions haven't occurred since ~ 40 ka (as summarised nicely on Ln. 183-184). The argument is made subsequently that existing climate data (ice core temperatures, Chironomid-inferred temperatures) show the only likely warm period prior to ~ 40 ka is the last interglacial, but how reliable are those indirect proxies as the basis for such a potentially ground-breaking conclusion? Recently, ice core oxygen isotope ratios have recently been described as reflecting winter temperature in Greenland (Buizert et al., 2014), not summer temperature, while

both ice core and chironomid-inferred temperatures appear to conflict with glacier-based reconstructions in the North Atlantic (Denton et al., 2005; Levy et al., 2016; Bromley et al., 2018). Until these discrepancies are reconciled sufficiently, I'd be highly wary of basing such a grand glacial interpretation on these less-direct proxies and would avoid like the plague use of such language as 'confirm' (Ln. 243) in relation to this conclusion.

The in situ ^{14}C measurements on bedrock are an exciting extension from the previously published work and, as stated above, form a logical progression to the plant ^{14}C analyses. These measurements indicate small concentrations of cosmogenic ^{14}C , which we are told represent limited exposure at these sites. I'm keen to know how these concentrations compare to the process blanks but can find no mention of the latter in the manuscript and so am unable to assess independently how 'low' some of these values truly are. Am I missing something here, or is this information that would normally be included?

Staying with the in situ ^{14}C , support for the principal finding of continual ice cover since 115 ka comes from the modelled Holocene ice thickness values. While I applaud the authors' coverage of this process, I am concerned that some of the assumptions used to calculate this parameter are dangerously indirect. Am I correct in thinking that an ice thickness 'record' for Arctic Canada has been estimated from the NGRIP oxygen isotope record (Ln. 136)? If so, that should come with a very clear statement of limitation, since a robust link between an isotopic record from Greenland and ice thickness (be that in Greenland or farther afield) does not, at least to my knowledge, exist. Furthermore, for some of these exposure/burial scenarios to reflect continuous Holocene cover, it would appear (e.g., in the caption for Table 2) that ice thickness of < 2 m is required for the duration of the Holocene thermal maximum. Assuming I am interpreting this assertion correctly, the dynamic nature of ice pretty much precludes such long-term (centennial or millennial) persistence of such a thin layer of ice on the landscape: glaciers grow and shrink, but there's no analogue for stationary thin ice lasting very long. In the event that I have woefully misunderstood that argument, I would encourage the authors to clarify their meaning as it's not sufficiently clear.

Minor comments:

Ln. 56: What makes these ice caps 'unique'?

Lns. 146-147, 168: You refer to 'unrealistically thick Neoglacial ice', but according to whom? And to what? Don't assume your readership will take everything at face value.

Lns. 226-228: Please specify which ice caps you are referring to. Conceptually, this makes sense, but it needs elaboration.

References:

Miller et al., 2013. Unprecedented recent summer warmth in Arctic Canada. GRL 40, 5745-5751.

Denton et al., 2005. The role of seasonality in abrupt climate change. QSR 24, 1159–1182.

Levy et al., 2016. Coeval fluctuations of the Greenland Ice Sheet and a local glacier, central East Greenland, during late-glacial and early Holocene time. GRL 43, 1623–1631.

Bromley et al., 2018. Interstadial Rise and Younger Dryas Demise of Scotland's Last Ice Fields. P&P 33, 10.1002/2018PA003341.

Reviewer #2 (Remarks to the Author):

Review of:

Rapidly receding Arctic Canada glaciers

revealing landscapes ice-covered for 115,000 years

by Pendleton et al.

Nature Communications, June 2018

In this manuscript, the authors describe a large dataset of old (>40 ka) radiocarbon ages from dead plants emerging from beneath Baffin Island ice caps. They use these ages to make the inference that the ice caps are smaller now than they have been since the last interglacial period ~115ka. They support their conclusion with cosmogenic ¹⁴C concentrations from rock surfaces and numerical modeling of those concentrations, demonstrating that exposure during the Holocene was unlikely. They argue that the current rate of warming in the Arctic has led to warmer summers and hence smaller ice caps than even during the Holocene Thermal Maximum, and that Baffin will likely be ice-free in the coming centuries.

Overall, I would like to commend the authors on a job well done. The paper is clear, well-written, and presents a coherent and focused argument. The topic is provocative and will be of interest to a diverse audience. The authors do a nice job describing the implications of their findings and of communicating why the small, thin, high-elevation ice caps on Baffin so effectively record this information. The authors have identified an interesting, unique geomorphic system and have exploited it effectively to address an important question.

However, the paper could benefit from revision. In particular, I think the information could be made more accessible to the uninitiated reader, especially given the diverse audience of the journal, and especially given confusion between the two types of 14C. Below, I detail several big picture comments as well as many smaller comments.

Again, I commend the authors on an interesting study,

Lee Corbett

University of Vermont

Ashley.Corbett@uvm.edu

Big-picture comments:

Accessibility to a diverse audience: At present, there are quite a few concepts in the paper that I felt were insufficiently defined for the diverse audience of Nature Communications. I tried to note some of these areas in the minor comments, below. I understand this is a short format paper, but your audience will not be able to understand and interpret your work without some of this background. In many cases, a few helpful words or phrases could be added parenthetically to help uninitiated readers with certain terms or phrases. In particular, the concept of the 14C half-life (and how it results in non-finite ages) could be better defined, and the difference between plant and rock 14C could be clarified (the latter of which will likely be a sticking point for many readers, who will have never heard of in situ cosmogenic 14C). Some of this information appears in the methods, but I think that is too late, and some people may not read that section at all.

Active voice: This is a personal opinion, but I advocate strongly for using active voice. You have passive voice scattered throughout, and it makes the text wordier than it needs to be and harder to understand. Especially since this dataset is building upon the smaller dataset that was initially published, there were places where I wasn't sure whether you were referring to your own work or to others' work that had been done previously (e.g. lines 110-111). I would rather see the text phrased in a more direct way, and the extra space used for additional clarification of important concepts.

Larger significance: I think the end of the paper is effective, in that it builds to the take-home point that Baffin will likely be ice-free in the coming centuries. However, I would like to see another sentence or two about why the diverse readership of Nature should care about this. Are there sea level implications? Ecological implications? Cultural implications? A full discussion of this is beyond the scope of the paper, but a few big-picture links (perhaps in both the introduction and the conclusion) would help make your work relevant and exciting to more people.

Minor comments:

Lines 22, 50, 253, and throughout: The 14C terminology is going to be tricky here because you're looking at two different kinds of 14C. I would maybe avoid using the term "in situ" with regards to your plants, since you then define "in situ" 14C as the cosmogenic variety in rocks. I worry this will be a point of confusion for readers. What if you describe the plants as being in growth position rather than in situ?

Line 29: "Exposure" is vague. Are you implying Holocene exposure? Exposure since initial ice cap growth?

Line 40: I'm not sure "millennial" is the right word here; to me it implies thousands of years, but you're thinking an order of magnitude or two longer than that.

Lines 42-43: Your phrase "constrain the timing of past glacier expansion" is a little misleading here. You're doing a lot more than just that- add another phrase so that your goals seem more broadly focused.

Line 48: It's not clear what timescale you're talking about here. When you talk about emergence and reburial, are you pointing toward the HTM? If so, be specific.

Line 50: You might add a very short parenthetical definition of in situ cosmogenic 14C here. It will be confusing for people what the difference is, and at present readers have to wait quite a while to understand the method and how it is difference from plant 14C.

Line 57: Clarify for the reader what the range of ^{14}C dating is and how it is dictated. A short parenthetical would suffice, like “as controlled by the isotopes half-life of 5730 years”.

Line 60: Rephrase the last sentence to focus on what your data do say (“all have ages $>27\text{ ka}$ ”) rather than what they don’t say (“none have ages $<27\text{ ka}$ ”). There are numerous places in the text that seem to have unnecessary negatives like this, and it makes the writing harder to understand and also wordier.

Figure 1: In my PDF, it looks like the north arrow is overlapping with the “N”.

Table 1: I worry that the “ $>$ ” symbol will be lost on readers not familiar with radioactive isotopes. I suggest adding a sentence into the table caption and/or text describing that non-finite ages arise when there is so little of the isotope left that it becomes non-detectable.

Line 99: Define what the “lifespan” is and what dictates it. Although many readers are probably familiar with ^{14}C , they may not have thought about its long-term life, i.e. what happens numerous half-lives out.

Lines 99-103: Define briefly what spallation and muogenic production are, i.e. “There are two different pathways by which cosmogenic isotopes can be produced...”. One sentence or a parenthetical is fine, but at present these terms won’t be accessible to the majority of readers.

Line 106: I would clarify for the reader here that the inventories are zero because all pre-LGM ^{14}C has decayed away.

Line 108: Emphasize again for the reader why thickness matters, something like “...ice thickness histories after $\sim 15\text{ ka}$, which dictated the amount of shielding and hence ^{14}C production”. I think this concept won’t be very familiar to readers, so anything you can do to help them along will be beneficial.

Line 110-111: You can get rid of the first sentence of the paragraph, it’s repetitive with what you have above; just present the data.

Lines 115-117: I'm not sure how useful this comparison is. I know you have it there for context, but a site that has been continuously exposed for 13ka is total apples and oranges from what you're doing. The extra detail may be confusing and misleading.

Lines 175-176: I'm having a hard time understanding why the earlier exposure possibility is incompatible with your observed concentrations. Describe?

Line 185: "Under" doesn't seem like the right word.

Lines 194-195: The double negative is confusing; reword.

Lines 196-198: I think I understand what you're getting at here, but it's not worded very clearly. I think all the negatives are tripping me up. Reword to focus on what does happen and how it impacted your samples.

Figure 4: This is really well done and useful. Clarify in the caption that the ages in panel F represent what you might expect for plant 14C (as opposed to rock 14C).

Lines 222-223: What is the "Late Holocene"? I'd prefer to see an age here, or tie it in to an event that more people are familiar with, e.g. the Little Ice Age. Otherwise, it's unconstrained in time.

Lines 232 and 246: Beware of the word "unprecedented" here; I suggest avoiding it, especially in the section heading where you can't clarify/qualify. I would use "the warmest conditions since the last interglacial period" or somesuch to avoid misconceptions. This word could really trip up a less knowledgeable audience and any media that may arise from the publication.

Line 234: I don't really like "return" here, it seems awkward. I think it is used a couple other times too, and seems sort of inanimate, like the plants dated themselves.

Line 280: Specify which primary standard the AMS analyses were normalized to and what its assumed value is.

Responses to Reviewer Remarks (annotated in red text)

Reviewer #1 (Remarks to the Author):

Review of Pendleton et al., ‘Rapidly receding Arctic Canada glaciers revealing landscapes ice-covered for 115,000 years’

The paper describes a glacial-chronologic approach to assessing modern deglaciation in Arctic Canada in the context of past glacial change, and builds on similar work by several of the authors (e.g., Miller et al., 2013). In so doing, the authors present a large radiocarbon dataset and a smaller in situ ^{14}C record, ultimately arriving at the conclusion that ice caps in this part of the Arctic have not been as small as present since the last interglacial, circa 115 ka. Independent of that principal finding, this research represents a large body of work and will make a valuable contribution to the palaeoclimate community’s understanding of natural boundary conditions for climate: How fast can our climate system fluctuate between states, and how has temperature evolved over the preceding cold and warm periods? At face value, this paper provides a truly tantalising glimpse of the impacts of abrupt climate shifts.

Methodologically, the plant ^{14}C chronology is a tried and tested (if not particularly novel) approach and the results, showing modern ice extent is unprecedented since at least 40 ka (or the limit of the radiocarbon method), appears to be robust. The subsequent measurement of in situ cosmogenic ^{14}C in bedrock represents a logical progression from the plant data and is indeed a novel element of the paper (more on that below). Moreover, the potential mechanisms by which retreat of these pedestal ice caps might be revealing landscapes today that were buried during the Holocene thermal maximum is well-illustrated in Figure 4. Beyond the plant-based data set, however, I feel the arguments for continuous ice cover since the last interglacial are overstepping the limits of the multi-proxy record, for reasons I hope to outline below.

Firstly, the gap between ~ 40 ka (beyond which the ^{14}C method is unreliable) and the last interglacial constitutes a period of approximately 70,000 years during which climate varied considerably. While the authors might indeed be correct in their inference that ice cover was continuous since 115 ka, there is no way to establish this possibility with a minimum-limiting data set such as this. All they can say with certainty is that ice-free conditions haven’t occurred since ~ 40 ka (as summarised nicely on Ln. 183-184).

We agree with the Reviewer here. Our data can only speak to the past ~ 40 ka. Text throughout the manuscript has been fine-tuned to adhere to this.

The argument is made subsequently that existing climate data (ice core temperatures, Chironomid-inferred temperatures) show the only likely warm period prior to ~ 40 ka is the last interglacial, but how reliable are those indirect proxies as the basis for such a potentially ground-breaking conclusion? Recently, ice core oxygen isotope ratios have recently been described as reflecting winter temperature in Greenland (Buizert et al., 2014), not summer temperature, while both ice core and chironomid-inferred temperatures appear to conflict with glacier-based reconstructions in the North Atlantic (Denton et al., 2005; Levy et al., 2016; Bromley et al., 2018). Until these discrepancies are reconciled sufficiently, I’d be highly wary of basing such a

grand glacial interpretation on these less-direct proxies and would avoid like the plague use of such language as ‘confirm’ (Ln. 243) in relation to this conclusion.

The Reviewer is correct that we must be careful with our language with regards to conclusions based on our presented data (our data presents a strong case for continuous ice cover since ~40 ka, but does not tell us anything about ice cover between 40-115 ka). However we stand by our speculation that these landscapes have likely been ice covered since the LIG based on temperature proxies from NGRIP and Baffin Lake chironomids. While there is significant variability in both records throughout the last glacial period, the LIG is the next most recent time when inferred temperatures consistently reach Holocene levels, hence our inference that our study locations have been ice covered since the LIG. Additionally, having two independent proxy records show similar trends strengthens this speculation.

The Reviewer also presents some concern of using these proxy records over glacier based records. However, the records they suggest are from farther afield than the temperature proxy records used here (not to mention that glaciers have an ice dynamic component that can complicate climate interpretations from paleo-glacial dimensions). More importantly, the records they cite all post-date our >40 ka plant radiocarbon ages. Since our plant (and *in situ* ^{14}C) together suggest continuous ice cover since ~40 ka, implying that, over average, temperatures did not reach modern levels over that time period. Thus, the logical next step is to locate the time period *prior* to ~40 ka where temperatures consistently reached Holocene levels, which is the LIG.

We reiterate that while our data itself does not speak to ice coverage between ~40-115 ka, it does suggest continuous ice cover for the past ~40 ka, strengthening the case for continuous ice cover since the LIG.

The *in situ* ^{14}C measurements on bedrock are an exciting extension from the previously published work and, as stated above, form a logical progression to the plant ^{14}C analyses. These measurements indicate small concentrations of cosmogenic ^{14}C , which we are told represent limited exposure at these sites. I’m keen to know how these concentrations compare to the process blanks but can find no mention of the latter in the manuscript and so am unable to assess independently how ‘low’ some of these values truly are. Am I missing something here, or is this information that would normally be included?

The Reviewer brings up valid points concerning the reporting of process blanks and uncertainty. Unlike other nuclide analyses (e.g., ^{10}Be , ^{26}Al), sources of contamination (e.g., atmospheric or inorganic ^{14}C) during AMS preparation require much stricter protocols. Realizing the importance of the process blank (especially for low concentration analyses), PRIME lab utilizes a protocol of multiple blank measurements to maximize precision, lower the detection limit, and improve confidence in the blank correction^{1,2}. In the case of this study, process blank ^{14}C concentrations are in general equal to or greater than the final measurement, which suggests a low level sample concentration. However, the recent improvements in the quantification of the blank gives us confidence in the final reported concentrations. Individual process blank values and discussion of detection limits have been added to the supplemental information.

Furthermore, when compared to other *in situ* ^{14}C analyses of surfaces with known ice cover histories on Baffin Island, the concentrations reported here are, in general, significantly lower. These comparisons provide context for the samples reported here and suggest significantly more burial than these other locations.

Staying with the *in situ* ^{14}C , support for the principal finding of continual ice cover since 115 ka comes from the modelled Holocene ice thickness values. While I applaud the authors' coverage of this process, I am concerned that some of the assumptions used to calculate this parameter are dangerously indirect. Am I correct in thinking that an ice thickness 'record' for Arctic Canada has been estimated from the NGRIP oxygen isotope record (Ln. 136)? If so, that should come with a very clear statement of limitation, since a robust link between an isotopic record from Greenland and ice thickness (be that in Greenland or farther afield) does not, at least to my knowledge, exist.

The Reviewer keys into one of the main model assumptions made in this study. The authors agree that use of the NGRIP record only provides an approximation of the ice thickness history at our field sites (especially given the distance between Baffin and Greenland), however, it is the best available record to approximate the ice thickness history. Text clarifying the use of the NGRIP record has been added to the manuscript.

The Authors would also mention that given the short half-life of ^{14}C , any *in situ* ^{14}C produced prior to ~40 ka has decayed away, meaning that 1) the ice cover history prior to ~40 ka has little to no influence on the modern *in situ* ^{14}C , and 2) ice cover *since* ~40 ka is the primary driver of the observed *in situ* ^{14}C concentration. Text to this effect has also been added.

Furthermore, for some of these exposure/burial scenarios to reflect continuous Holocene cover, it would appear (e.g., in the caption for Table 2) that ice thickness of < 2 m is required for the duration of the Holocene thermal maximum. Assuming I am interpreting this assertion correctly, the dynamic nature of ice pretty much precludes such long-term (centennial or millennial) persistence of such a thin layer of ice on the landscape: glaciers grow and shrink, but there's no analogue for stationary thin ice lasting very long. In the event that I have woefully misunderstood that argument, I would encourage the authors to clarify their meaning as it's not sufficiently clear.

The Reviewer is correct here; for an HTM between 8.6 – 4.9 ka, the model requires some locations to maintain an HTM ice cover of only ~2 m, which is unlikely given the variable nature of glacier thickness. Upon review, of these thin ice covers, we have also realized that our currently imposed linear thinning of ice following the LGM does not adequately capture the likely ice cover during deglaciation. To improve this part of the model we have updated our ice history to follow the melt record and modeled ice thinning from Agassiz Ice Cap³, which calls for rapid thinning before ~9 ka, followed by significantly slowed thinning from ~9 to ~5 ka. Use of this record to drive our deglacial ice history allows us to remove the use of imposed periods of constant ice thickness, and we think more accurately represents the thinning of ice during deglaciation.

Rerunning of the simulations with this improved ice cover history yields very similar results to the previous iteration. The three sites with the highest *in situ* ^{14}C still require thin ice during the middle Holocene (1-2m), a thickness that is likely untenable for several thousand years. However, the three sites with the highest *in situ* ^{14}C require ice to be ≥ 6 m during the middle Holocene.

Moreover, we have added text to the modeling section emphasizing that while the simulations show that continuous Holocene coverage is possible under realistic ice covers, the simulations alone do not rule out the possibility of Holocene exposure. It is the combination of the *in situ* ^{14}C and the plant ^{14}C that presents the strongest case for continuous Holocene ice cover. The language in the Conclusions section has also been altered to emphasize these points.

Minor comments:

Ln. 56: What makes these ice caps ‘unique’?

The author’s intended to convey that 30 *different* ice caps were sampled. Text clarified.

Lns. 146-147, 168: You refer to ‘unrealistically thick Neoglacial ice’, but according to whom? And to what? Don’t assume your readership will take everything at face value.

The Reviewer makes an excellent point. We categorize ‘unrealistically thick’ ice at >70 m, which is based on the topographic setting of the ice caps and inherent ice rheology (discussed later in the manuscript). Text and citations added to clarify this.

Lns. 226-228: Please specify which ice caps you are referring to. Conceptually, this makes sense, but it needs elaboration.

Specific ice cap referred to in text.

References:

Miller et al., 2013. Unprecedented recent summer warmth in Arctic Canada. GRL 40, 5745-5751.

Denton et al., 2005. The role of seasonality in abrupt climate change. QSR 24, 1159–1182.

Levy et al., 2016. Coeval fluctuations of the Greenland Ice Sheet and a local glacier, central East Greenland, during late-glacial and early Holocene time. GRL 43, 1623–1631.

Bromley et al., 2018. Interstadial Rise and Younger Dryas Demise of Scotland’s Last Ice Fields. P&P 33, 10.1002/2018PA003341.

Reviewer #2 (Remarks to the Author):

Review of: Rapidly receding Arctic Canada glaciers revealing landscapes ice-covered for 115,000 years

by Pendleton et al.

Nature Communications, June 2018

In this manuscript, the authors describe a large dataset of old (>40 ka) radiocarbon ages from dead plants emerging from beneath Baffin Island ice caps. They use these ages to make the inference that the ice caps are smaller now than they have been since the last interglacial period ~115ka. They support their conclusion with cosmogenic ¹⁴C concentrations from rock surfaces and numerical modeling of those concentrations, demonstrating that exposure during the Holocene was unlikely. They argue that the current rate of warming in the Arctic has led to warmer summers and hence smaller ice caps than even during the Holocene Thermal Maximum, and that Baffin will likely be ice-free in the coming centuries.

Overall, I would like to commend the authors on a job well done. The paper is clear, well-written, and presents a coherent and focused argument. The topic is provocative and will be of interest to a diverse audience. The authors do a nice job describing the implications of their findings and of communicating why the small, thin, high-elevation ice caps on Baffin so effectively record this information. The authors have identified an interesting, unique geomorphic system and have exploited it effectively to address an important question.

However, the paper could benefit from revision. In particular, I think the information could be made more accessible to the uninitiated reader, especially given the diverse audience of the journal, and especially given confusion between the two types of ^{14}C . Below, I detail several big picture comments as well as many smaller comments.

Again, I commend the authors on an interesting study,

Lee Corbett
University of Vermont
Ashley.Corbett@uvm.edu

Big-picture comments:

Accessibility to a diverse audience: At present, there are quite a few concepts in the paper that I felt were insufficiently defined for the diverse audience of Nature Communications. I tried to note some of these areas in the minor comments, below. I understand this is a short format paper, but your audience will not be able to understand and interpret your work without some of this background. In many cases, a few helpful words or phrases could be added parenthetically to help uninitiated readers with certain terms or phrases. In particular, the concept of the ^{14}C half-life (and how it results in non-finite ages) could be better defined, and the difference between plant and rock ^{14}C could be clarified (the latter of which will likely be a sticking point for many readers, who will have never heard of in situ cosmogenic ^{14}C). Some of this information appears in the methods, but I think that is too late, and some people may not read that section at all.

The Authors agree with the Reviewer here; a clearer definition of what a non-finite radiocarbon age is has been added to the text. The language regarding the difference between plant and rock ^{14}C has also been clarified.

Active voice: This is a personal opinion, but I advocate strongly for using active voice. You have passive voice scattered throughout, and it makes the text wordier than it needs to be and harder to understand. Especially since this dataset is building upon the smaller dataset that was initially published, there were places where I wasn't sure whether you were referring to your own work or to others' work that had been done previously (e.g. lines 110-111). I would rather see the text phrased in a more direct way, and the extra space used for additional clarification of important concepts.

We thank the Reviewer for their helpful comments regarding passive vs active voice. Changes have been made throughout the manuscript to more directly convey information and improve clarity (see below for specific changes).

Larger significance: I think the end of the paper is effective, in that it builds to the take-home point that Baffin will likely be ice-free in the coming centuries. However, I would like to see another sentence or two about why the diverse readership of Nature should care about this. Are there sea level implications? Ecological implications? Cultural implications? A full discussion of

this is beyond the scope of the paper, but a few big-picture links (perhaps in both the introduction and the conclusion) would help make your work relevant and exciting to more people.

This is an important comment regarding the broad importance of this work. We agree with the Reviewer that a lengthy discussion is beyond the scope of this paper (and word limit) we have added some bigger-picture comments that will hopefully connect with a broader readership and promote future work regarding persisting unknowns.

Minor comments:

Lines 22, 50, 253, and throughout: The ^{14}C terminology is going to be tricky here because you're looking at two different kinds of ^{14}C . I would maybe avoid using the term "in situ" with regards to your plants, since you then define "in situ" ^{14}C as the cosmogenic variety in rocks. I worry this will be a point of confusion for readers. What if you describe the plants as being in growth position rather than in situ?

We agree with the Reviewer: changes have been made to specify the types of radiocarbon used in this study. The ages of plants ingrowth position are referred to as 'plant ^{14}C ' while the cosmogenic ^{14}C measured in rock surfaces is '*in situ* ^{14}C '.

Line 29: "Exposure" is vague. Are you implying Holocene exposure? Exposure since initial ice cap growth?

'...Holocene exposure...' specified.

Line 40: I'm not sure "millennial" is the right word here; to me it implies thousands of years, but you're thinking an order of magnitude or two longer than that.

'...multi-millennial...' used to specify the longer timescale.

Lines 42-43: Your phrase "constrain the timing of past glacier expansion" is a little misleading here. You're doing a lot more than just that- add another phrase so that your goals seem more broadly focused.

Additional objective phrase added. ('... and provide context for current warming.')

Line 48: It's not clear what timescale you're talking about here. When you talk about emergence and reburial, are you pointing toward the HTM? If so, be specific.

HTM specified.

Line 50: You might add a very short parenthetical definition of in situ cosmogenic ^{14}C here. It will be confusing for people what the difference is, and at present readers have to wait quite a while to understand the method and how it is difference from plant ^{14}C .

Short definition of *in situ* ^{14}C inserted.

Line 57: Clarify for the reader what the range of ^{14}C dating is and how it is dictated. A short parenthetical would suffice, like "as controlled by the isotopes half-life of 5730 years".

Agreed, text added.

Line 60: Rephrase the last sentence to focus on what your data do say (“all have ages >27 ka”) rather than what they don’t say (“none have ages <27 ka”). There are numerous places in the text that seem to have unnecessary negatives like this, and it makes the writing harder to understand and also wordier.

Manuscript reviewed for unnecessary negatives and converted to positives where appropriate.

Figure 1: In my PDF, it looks like the north arrow is overlapping with the “N”.

North arrow and ‘N’ separated.

Table 1: I worry that the “>” symbol will be lost on readers not familiar with radioactive isotopes. I suggest adding a sentence into the table caption and/or text describing that non-finite ages arise when there is so little of the isotope left that it becomes non-detectable.

Caption text changed.

Line 99: Define what the “lifespan” is and what dictates it. Although many readers are probably familiar with ^{14}C , they may not have thought about its long-term life, i.e. what happens numerous half-lives out.

Agreed, text added to clarify.

Lines 99-103: Define briefly what spallation and muogenic production are, i.e. “There are two different pathways by which cosmogenic isotopes can be produced...”. One sentence or a parenthetical is fine, but at present these terms won’t be accessible to the majority of readers.

Production pathways definitions added.

Line 106: I would clarify for the reader here that the inventories are zero because all pre-LGM ^{14}C has decayed away.

Text clarified.

Line 108: Emphasize again for the reader why thickness matters, something like “...ice thickness histories after ~15 ka, which dictated the amount of shielding and hence ^{14}C production”. I think this concept won’t be very familiar to readers, so anything you can do to help them along will be beneficial.

Agreed; text added.

Line 110-111: You can get rid of the first sentence of the paragraph, it’s repetitive with what you have above; just present the data.

Sentence removed.

Lines 115-117: I’m not sure how useful this comparison is. I know you have it there for context, but a site that has been continuously exposed for 13ka is total apples and oranges from what you’re doing. The extra detail may be confusing and misleading.

The Authors believe the comparisons to other measured *in situ* ^{14}C inventories on Baffin Island with known histories provides context for our generally lower inventories. However, we see the Reviewers point about confusion. If the editor deems it appropriate, we can remove the latter two comparisons, leaving the continuously exposed summit reported here as the sole

comparison.

Lines 175-176: I'm having a hard time understanding why the earlier exposure possibility is incompatible with your observed concentrations. Describe?

Earlier exposure scenario results added to Table 2. Most of measured *in situ* 14C inventories cannot be reproduced realistic ice cover following exposure from 11-9 ka.

Line 185: "Under" doesn't seem like the right word.

Sentence reworded.

Lines 194-195: The double negative is confusing; reword.

Sentence reworded.

Lines 196-198: I think I understand what you're getting at here, but it's not worded very clearly. I think all the negatives are tripping me up. Reword to focus on what does happen and how it impacted your samples.

Sentence altered to clarify.

Figure 4: This is really well done and useful. Clarify in the caption that the ages in panel F represent what you might expect for plant 14C (as opposed to rock 14C).

Clarified in Figure 4 caption.

Lines 222-223: What is the "Late Holocene"? I'd prefer to see an age here, or tie it in to an event that more people are familiar with, e.g. the Little Ice Age. Otherwise, it's unconstrained in time.

Timing clarified in text.

Lines 232 and 246: Beware of the word "unprecedented" here; I suggest avoiding it, especially in the section heading where you can't clarify/qualify. I would use "the warmest conditions since the last interglacial period" or something to avoid misconceptions. This word could really trip up a less knowledgeable audience and any media that may arise from the publication.

Word 'unprecedented' removed and sentence reworked according to Reviewers recommendations.

Line 234: I don't really like "return" here, it seems awkward. I think it is used a couple other times too, and seems sort of inanimate, like the plants dated themselves.

Replaced 'return' with 'yield'.

Line 280: Specify which primary standard the AMS analyses were normalized to and what its assumed value is.

References cited by Authors

1. Lifton, N., Jull, A. & Quade, J. A new extraction technique and production rate estimate for in situ cosmogenic ^{14}C in quartz. *Geochim. Cosmochim. Acta* **65**, 1953–1969 (2001).
2. Lifton, N., Goehring, B., Wilson, J., Kubley, T. & Caffee, M. Progress in automated extraction and purification of in situ ^{14}C from quartz: Results from the Purdue in situ ^{14}C laboratory. *Nucl. Instruments Methods Phys. Res. Sect. B Beam Interact. with Mater. Atoms* **361**, 381–386 (2015).
3. Lecavalier, B. S. *et al.* High Arctic Holocene temperature record from the Agassiz ice cap and Greenland ice sheet evolution. *Proc. Natl. Acad. Sci.* **114**, 5952–5957 (2017).
4. Axford, Y. *et al.* Chironomids record terrestrial temperature changes throughout Arctic interglacials of the past 200,000 yr. *Geol. Soc. Am. Bull.* **123**, 1275–1287 (2011).

REVIEWERS' COMMENTS:

Reviewer #1 (Remarks to the Author):

The authors have addressed in detail the majority of the concerns I raised earlier and produced a revised manuscript that is, to my eyes, as transparent as possible. Therefore, readers will be in a position to decide for themselves whether, e.g., NGRIP oxygen isotopes are an accurate measure of Arctic mean temperature, and the ramifications for this study. I also appreciate the refined explanation of Holocene ice thickness, which goes a long way to making the paper more credible. All in all, I congratulate the authors on a job well done and look forward to seeing the manuscript in print.

Gordon Bromley

Reviewer #2 (Remarks to the Author):

Second review of:

Rapidly receding Arctic Canada glaciers

revealing landscapes ice-covered for 115,000 years

by Pendleton et al.

Nature Communications, September 2018

In this manuscript, the authors describe a large dataset of old (>40 ka) radiocarbon ages from dead plants emerging from beneath Baffin Island ice caps. They use these ages to make the inference that the ice caps are smaller now than they have been since the last interglacial period ~115ka.

In this revised version of the paper, the authors have clarified certain areas of the text that I had worried might be confusing for the reader. They have also broadened the implications of the paper in order to make it more accessible and interesting to a diverse audience.

In my opinion, the paper is now ready for publication. This is a very interesting study with provocative, potentially wide-reaching results. I look forward to seeing where it leads.

Lee Corbett

University of Vermont

Ashley.Corbett@uvm.edu

REVIEWERS' COMMENTS:

****The reviewer comments below detail no comments or corrections that need to be addressed in this resubmission***

Reviewer #1 (Remarks to the Author):

The authors have addressed in detail the majority of the concerns I raised earlier and produced a revised manuscript that is, to my eyes, as transparent as possible. Therefore, readers will be in a position to decide for themselves whether, e.g., NGRIP oxygen isotopes are an accurate measure of Arctic mean temperature, and the ramifications for this study. I also appreciate the refined explanation of Holocene ice thickness, which goes a long way to making the paper more credible. All in all, I congratulate the authors on a job well done and look forward to seeing the manuscript in print.

Gordon Bromley

Reviewer #2 (Remarks to the Author):

Second review of:
Rapidly receding Arctic Canada glaciers
revealing landscapes ice-covered for 115,000 years
by Pendleton et al.
Nature Communications, September 2018

In this manuscript, the authors describe a large dataset of old (>40 ka) radiocarbon ages from dead plants emerging from beneath Baffin Island ice caps. They use these ages to make the inference that the ice caps are smaller now than they have been since the last interglacial period ~115ka.

In this revised version of the paper, the authors have clarified certain areas of the text that I had worried might be confusing for the reader. They have also broadened the implications of the paper in order to make it more accessible and interesting to a diverse audience.

In my opinion, the paper is now ready for publication. This is a very interesting study with provocative, potentially wide-reaching results. I look forward to seeing where it leads.

Lee Corbett
University of Vermont
Ashley.Corbett@uvm.edu